# Sorting out assortativity: When can we assess the contributions of different population groups to epidemic transmission?

Cyril Geismar[1,2]*, Peter J. White[1,2,3], Anne Cori[1,4�], Thibaut Jombart[1,2�]

**1** MRC Centre for Global Infectious Disease Analysis, Imperial College School of Public Health, London, United Kingdom, **2** NIHR Health Protection Research Unit in Modelling and Health Economics, Imperial College School of Public Health, London, United Kingdom, **3** Modelling & Economics Unit, UK Health Security Agency, London, United Kingdom, **4** Abdul Latif Jameel Institute for Disease and Emergency Analytics, Imperial College School of Public Health, London, United Kingdom

☉ These authors contributed equally to this work.
* c.geismar21@imperial.ac.uk

**Data Availability Statement:** The R package developed to estimate transmission assortativity from transmission chain data can be accessed at:

## Abstract

Characterising the transmission dynamics between various population groups is critical for implementing effective outbreak control measures whilst minimising financial costs and societal disruption. While recent technological and methodological advances have made individual-level transmission chain data increasingly available, it remains unclear how effectively this data can inform group-level transmission patterns, particularly in small, rapidly saturating outbreak settings. We introduce a novel framework that leverages transmission chain data to estimate group transmission assortativity; this quantifies the extent to which individuals transmit within their own group compared to others. Through extensive simulations mimicking nosocomial outbreaks, we assessed the conditions under which our estimator performs effectively and established guidelines for minimal data requirements in small outbreak settings where saturation may occur rapidly. Notably, we demonstrate that detecting and quantifying transmission assortativity is most reliable when at least 30 cases have been observed in each group, before reaching their respective epidemic peaks.

## Introduction

Understanding the heterogeneous contributions of population groups to disease transmission is crucial for developing effective targeted interventions whilst minimising financial costs and societal disruption. Individuals can be categorised by age [1–3], occupation [4], vaccination status [5], sexual preferences [6–8] and other characteristics relevant to the disease context [9, 10]. These group dynamics are not only determined by distinct contact patterns, as revealed by large-scale contact surveys studies, such as POLYMOD [11] and CoMix [12], but also by varying infectiousness and susceptibility levels [13, 14]. Heterogeneous transmission is particularly salient in healthcare settings, where the confined hospital environment and frequent interactions between healthcare workers (HCWs) and vulnerable patients in various wards [15] create a complex transmission landscape, potentially increasing the risk of infection compared to the general population [16, 17]. Nosocomial outbreaks not only pose significant risks to global

https://github.com/CyGei/linktree. Additionally, an R package for simulating outbreaks involving multiple groups with various transmission assortativity coefficients is available at: https://github.com/CyGei/o2groups. The code used for the analysis presented in this manuscript can be found at: https://github.com/CyGei/o2groups-analysis. Package and analysis code have been archived on Zenodo (analysis: https://zenodo.org/doi/10.5281/zenodo.10616176, package: https://zenodo.org/doi/10.5281/zenodo.10616155).

**Funding:** CG is supported by a PhD studentship at Imperial College London funded by the National Institute for Health Research (NIHR) Health Protection Research Unit (HPRU) in Modelling and Health Economics, which is a partnership between the UK Health Security Agency (UKHSA), Imperial College London, and the London School of Hygiene & Tropical Medicine (grant code NIHR200908). AC, PJW, TJ are supported by the HPRU in Modelling and Health Economics. This work was supported by the UK Medical Research Council (MRC) Centre for Global Infectious Disease Analysis (grant number MR/X020258/1); this award comes under the Global Health EDCTP3 Joint Undertaking. The funders had no role in study design, data collection and analysis, decision to publish, or preparation of the manuscript.

**Competing interests:** The authors have declared that no competing interests exist.

healthcare systems but also worsen patient outcomes and mortality while straining hospital resources and operational capacities [18]. Between March and July 2020, Evans *et al.* estimated that nosocomial transmission in the UK accounted for 20% and 73% of SARS-CoV-2 infections amongst inpatients and HCWs, respectively [19]. Cooper *et al.* further estimated from June 2020 to March 2021 that 1–2% of hospital admissions in England likely acquired SARS-CoV-2 while hospitalised, primarily driven by patient-to-patient transmission [18].

To characterise these group-specific transmission dynamics, modellers have traditionally relied on contact survey data [11, 12], combined with information about the relative infectiousness and/or susceptibility of each group (*e.g.* obtained from epidemiological or serological investigations) [13, 14]. However, this survey data can be biased, have limited sample size or representativeness, and may not be generalisable across different epidemic contexts [20].

Worby *et al.* introduced the Relative Risk (RR) statistic as an alternative approach to evaluate group contributions to epidemic transmission [1, 2]. This metric compares the proportion of cases attributed to a particular group before and after the epidemic peak relative to the total number of cases, offering insights into the group's relative depletion of susceptibles during the epidemic's ascent. While the RR statistic may be useful to determine which group to prioritise for vaccination in large outbreaks with synchronous peaks, it has significant limitations in smaller settings such as nosocomial outbreaks involving rapidly spreading respiratory pathogens such as coronaviruses, influenza, respiratory syncytial virus, or rhinovirus. These outbreaks often feature a small number of cases, variations in group sizes, numbers of imports, contacts and transmission patterns, resulting in asynchronous epidemic peaks. In these settings, Worby's premise that the depletion of susceptible individuals within a group reflects its role in driving the epidemic may not hold.

Given the constraints of conventional methods, innovative approaches are essential to address the challenges posed by nosocomial outbreaks. Healthcare settings are particularly suited for employing advanced outbreak reconstruction tools due to the relatively small size of nosocomial outbreaks, regular surveillance and data collection, and access to whole genome sequencing [21–26]. Such tools typically leverage pathogen genetic sequence data, symptom onset or test collection dates, and contact data within a Bayesian framework to probabilistically reconstruct transmission events, generating posterior sets of transmission chains [27–29]. Research has shown that a significant proportion of HCW SARS-CoV-2 infections are often attributable to their colleagues [21–25], whereas Cook *et al.* found that patient-to-patient and patient-to-HCW transmissions were comparatively more common [26]. However, these studies typically report the proportion of specific transmission types relative to the total number of transmissions (*e.g.*, 70% of transmissions were patient-to-patient). Without comparing these observations to expected frequencies, these approaches do not elucidate the underlying transmission dynamics that drive the outbreak.

To address this limitation, Abbas *et al.* developed a statistical test to detect non-random group transmission patterns using the outputs of Bayesian outbreak reconstruction tools [21, 23]. From the reconstructed chains of transmission, the authors estimated the proportion of infections caused by each case type and compared it to an expected proportion based on that type's prevalence amongst all cases [21, 23]. In a nosocomial SARS-COV-2 outbreak at a rehabilitation clinic, they identified that HCWs transmitted more frequently than expected [21]. However, the method's reliance on case prevalence alone neglects the process from exposure to infection and overlooks factors that shape the observed epidemic. Varying contact rates, susceptibility, and distinct mixing patterns can lead to a disproportionate number of a given case type from the outset, making prevalence-based expectations potentially misleading.

Furthermore, existing methods often fail to account for the rapid depletion of susceptibles typical in small outbreaks, such as those in healthcare settings. This saturation effect can

significantly alter transmission patterns, as a saturated group cannot sustain further transmission within the group. Thus analysing proportions of transmission types across an entire outbreak, beyond the point of saturation, may not accurately reflect the underlying baseline transmission patterns.

This paper introduces a novel framework for evaluating transmission patterns amongst distinct groups during an outbreak, addressing the limitations of previous methods. Our approach quantifies group-specific transmission assortativity, from known or probabilistically reconstructed transmission chains, while accounting for group sizes. We evaluate the performance of our estimator through diverse simulations, mimicking nosocomial outbreaks where the populations are fully susceptible at the outbreak's onset. Our aim is to provide guidelines on the minimum data collection requirements and the optimal estimation timeframe, thereby informing IPC strategies in small outbreaks where rapid saturation occurs.

## Methods

### A new estimator of transmission assortativity

Assortativity has been amply described for social mixing patterns, with homogeneous mixing referring to random contacts between individuals, and heterogeneous mixing denoting interactions characterised by distinct (non-random) patterns depending on group memberships [13, 30]. Heterogeneous mixing can be either *assortative*, where individuals tend to interact more within their own group (*e.g.* social contacts by age [11, 31, 32]), or *disassortative*, where individuals interact preferentially with members of other groups (*e.g.* sexual contacts [33]). Here we use these definitions to characterise the patterns of transmission rather than contact. The resulting transmission patterns thus reflect not only mixing patterns but also differences in infectiousness and susceptibility amongst groups.

To quantify transmission assortativity, we examine the person-to-person transmission patterns. We consider $G$ groups of relative sizes $f_1, \ldots, f_G$ defined as:

$$f_a = \frac{N_a}{\sum_{g=1}^{G} N_g} \quad \forall a = 1, \ldots, G \tag{1}$$

where $N_a$ is the number of individuals in group $a$. We denote $\beta_{b \leftarrow a}$ the person-to-person transmission rate from an individual in group $a$ to an individual in group $b$, that is the force of infection that any one infected individual in group $a$ exerted on any one susceptible individual in group $b$. It follows that, in a fully susceptible population, the expected number of secondary cases in group $b$ generated by one infectious individual in group $a$ is proportional to $\beta_{b \leftarrow a} N_b \propto \beta_{b \leftarrow a} f_b$.

We make the following assumptions:

1. $\beta_{b \leftarrow a}$ is the same for all $b \neq a$, *i.e.*, $\beta_{b \leftarrow a} = \psi$ if $a \neq b$ (S3 in S1 File).

2. $\beta_{a \leftarrow a} = \gamma_a \psi$, where $\gamma_a$ is the assortativity coefficient for group $a$.

The assortativity coefficient, $\gamma_a$, is defined as the excess probability of a secondary infection taking place within group $a$ compared to random expectation. $\gamma$ values range from 0 (fully disassortative, *i.e.* no within-group transmissions) to $\infty$ (fully assortative, *i.e.* transmissions occur exclusively within the group), with 1 indicating homogeneous patterns. For instance, $\gamma_a = 2$ indicates that an infected individual from group $a$ is twice as likely to infect an individual from the same group compared to infecting an individual from another group. Conversely, a $\gamma_a$ of 1/2 means that an infected individual from group $a$ is twice as likely to infect an individual from another group compared to infecting an individual from the same group.

We derive $\pi_{b \leftarrow a}$, the proportion of secondary cases in group $b$ amongst those generated by an infectious individual in group $a$, where $a \neq b$, as:

$$\pi_{b \leftarrow a} = \frac{\beta_{b \leftarrow a} f_b}{\sum_{g=1}^{G} \beta_{g \leftarrow a} f_g} = \frac{\beta_{b \leftarrow a} f_b}{\sum_{\substack{g=1 \\ g \neq a}}^{G} \beta_{g \leftarrow a} f_g + \beta_{a \leftarrow a} f_a} = \frac{\beta_{b \leftarrow a} f_b}{\psi(1-f_a) + \gamma_a \psi f_a} = \frac{\psi f_b}{\psi(1-f_a) + \gamma_a \psi f_a}$$

$$= \frac{f_b}{(1-f_a) + \gamma_a f_a} \tag{2}$$

We derive $\pi_{a \leftarrow a}$, the proportion of secondary cases in group $a$ amongst those generated by an infectious individual from the same group, as:

$$\pi_{a \leftarrow a} = \frac{\beta_{a \leftarrow a} f_a}{\psi(1-f_a) + \gamma_a \psi f_a} = \frac{\gamma_a \psi f_a}{\psi(1-f_a) + \gamma_a \psi f_a} = \frac{\gamma_a f_a}{(1-f_a) + \gamma_a f_a} \tag{3}$$

We can obtain $\gamma_a$ by rewriting Eq 3 as:

$$\gamma_a = \frac{\pi_{a \leftarrow a} \cdot (1-f_a)}{f_a \cdot (1-\pi_{a \leftarrow a})} \tag{4}$$

The proportion of within-group transmission, $\pi_{a \leftarrow a}$ (Eq 3), can be directly calculated from known transmission chain data. It is calculated by dividing the number of observed within-group transmission pairs, $\tau_{a \leftarrow a}$, by the total number of transmissions originating from group $a$, $\tau_{. \leftarrow a}$. Thus the proportion, $\pi_{a \leftarrow a} = \tau_{a \leftarrow a}/\tau_{. \leftarrow a}$, ranges between 0 and 1 (included).

To simplify interpretation, we introduce a rescaled parameter $\delta$, ranging between -1 (fully disassortative) and 1 (fully assortative), with 0 corresponding to a homogeneous transmission pattern (Fig 1.1 in S1 File) such that:

$$\delta = \begin{cases} 1 & \text{if } \gamma = \infty \\ \dfrac{\gamma - 1}{\gamma + 1} & \text{if } \gamma \neq \infty \end{cases} \tag{5}$$

The formula for $\delta_a$ can thus be written as:

$$\delta_a = \frac{\frac{\tau_{a \leftarrow a}}{\tau_{. \leftarrow a}} - f_a}{\frac{\tau_{a \leftarrow a}}{\tau_{. \leftarrow a}} + f_a \left(1 - 2\frac{\tau_{a \leftarrow a}}{\tau_{. \leftarrow a}}\right)} \tag{6}$$

where:

- $\tau_{a \leftarrow a}$: represents the number of transmissions from group $a$ towards group $a$.

- $\tau_{. \leftarrow a}$: refers to the total number of transmissions emitted by group $a$.

- $\tau_{a \leftarrow a}/\tau_{. \leftarrow a}$: Denotes the proportion of within-group transmissions for group $a$ denoted as $\pi_{a \leftarrow a}$ in Eq 3.

- $f_a$: Represents the proportion of the total population that belongs to group $a$. It is a value between 0 and 1, exclusive of the endpoints as there must be more than 1 group in the population.

The relationship between $\delta_a$, $\pi_{a \leftarrow a}$ (i.e. $\tau_{a \leftarrow a}/\tau_{. \leftarrow a}$), and $f_a$ can is visually represented in supplementary Fig 1.2 in S1 File. We can obtain a confidence interval (CI) on $\pi_{a \leftarrow a}$ for various significance ($\alpha$) levels using the Clopper-Pearson binomial interval method [34] (S1.1 in S1 File). Feeding estimates of $\pi_{a \leftarrow a}$ into Eq 6 provides estimates of $\delta_a$ with confidence intervals (S1.1 in S1 File).

All our results are presented using $\delta$ rather than $\gamma$.

## Simulation study

We simulated small outbreaks under various contexts to assess the estimator's performance in scenarios relevant to person-to-person transmission of healthcare-acquired pathogens in a fully susceptible population. We constructed 10,000 sets of input parameters, referred to as 'scenarios', by randomly sampling parameters from pre-defined distributions (Section 1.2 and Fig 2 in S1 File). To account for stochasticity, we conducted 100 simulations for each unique scenario resulting in a total of 1,000,000 simulated outbreaks.

The simulation employed a discrete time branching process modelling individual infections spreading in successive generations. Simulations were specified with: i) group-level parameters including the size of each group, their assortativity coefficients ($\delta$), initial introductions, basic reproduction numbers ($R_0$) and ii) epidemic level parameters such as the number of groups, the pathogen generation time ($w$) and incubation period distributions (both assumed the same across groups). The simulation outputs a transmission tree that includes, for each infected individual, their symptom onset date, their group affiliation, and the id of their infector. Using this data for all infected individuals with symptoms up to time t, we calculated $\tau_{a \leftarrow a}$ (the number of within-group transmission pairs) and $\tau_{. \leftarrow a}$ (the total number of transmissions originating from each group). These values, along with the relative sizes of the groups ($f_a$), were input into Eq 6 to estimate the assortativity coefficients for each group.

In our branching process model, the force of infection (FOI) generated by individual $j$ from group $a$ at time $t$, towards the whole of group $b$ is defined as:

$$\lambda_{b \leftarrow a}^{j}(t) = w(t - s_a^j) R_{0a} \pi_{b \leftarrow a} \quad \begin{array}{l} \forall a, b = 1, \ldots, G \\ \forall j = 1, \ldots, N_a \end{array} \tag{7}$$

where:

- $s_a^j$ is the time of infection of individual $j$ in group $a$

- $R_{0a}$ is the basic reproduction number of individuals in group $a$

- $w$ is the probability mass function of the generation time distribution

The total FOI that all individuals in group $b$ collectively receive from all individuals across all groups at time $t$ is obtained as:

$$\lambda_b(t) = \sum_{a=1}^{G} \sum_{j=1}^{N_a} \lambda_{b \leftarrow a}^{j}(t) \ \ \forall b = 1, \ldots, G \tag{8}$$

Hence, the FOI that one individual from group $b$ is exposed to is $\frac{\lambda_b(t)}{N_b}$.

The probability of infection for each individual in group $b$ at time $t$ is then calculated as:

$$p_b(t) = 1 - e^{-\frac{\lambda_b(t)}{N_b}} \tag{9}$$

At time $t + 1$, the number of new cases in group $b$, $X_b(t + 1)$, is drawn from a binomial distribution:

$$X_b(t + 1) \sim \text{Binom}(S_b(t), p_b(t)) \tag{10}$$

where $S_b(t)$ is the number of susceptible individuals in group $b$ at time $t$.

New cases are allocated at random amongst susceptible individuals. The simulation progresses in discrete daily time steps for 365 days. Nearly all simulations (99.99%) finished with the last infection occurring before day 300. Note that we assume that individuals who have been infected become fully immune.

Assuming that $b^i$ (i$^{\text{th}}$ individual in group $b$) was infected at time $t+1$, their infector $\alpha_{b^i}$ is drawn across all infected individuals in all groups from a multinomial distribution with probabilities:

$$p(\alpha_{b^i} = a^j)(t+1) = \frac{\lambda^j_{b \leftarrow a}(t)}{\lambda_b(t)} \qquad (11)$$

Where $a^j$ is the $j^{th}$ individual in group $a$.

To assess the performance of our estimator, we computed 4 different performance metrics for each scenario:

- *Bias*: defined as the average difference between the true $\delta$ value and its estimate ($\hat{\delta}$) across 100 simulations. It is a measure of the estimator's systematic error and inaccuracy and should be close to 0. Bias is positive when $\delta$ is underestimated, indicating underestimation of assortativity or overestimation of disassortativity. Conversely, negative bias occurs when $\delta$ is overestimated, indicating overestimation of assortativity or underestimation of disassortativity.

- *Coverage (at significance level $\alpha$)*: defined as the proportion of simulations (out of 100) where the true $\delta$ value is within the estimated CI corresponding to $\alpha$. We evaluate 4 significance levels 0.05, 0.1, 0.25 and 0.5. Assessing coverage helps determine the reliability of the confidence intervals generated by the estimator. Coverage should approximate 1-$\alpha$, and the coverage error, which measures the deviation from this target, should be close to 0. A positive coverage error suggests underestimation of uncertainty, while a negative coverage error indicates overestimation.

- *Sensitivity* (true positive rate): defined as the proportion of simulations (out of 100) where the estimator correctly identifies a significant assortative or disassortative effect (*i.e.* the $\hat{\delta}$ CI doesn't contain 0). Sensitivity should be close to 1 (100%).

- *Specificity* (true negative rate): defined as the proportion of simulations (out of 100) where the estimator correctly identifies no significant assortative or disassortative effect (*i.e.* the $\hat{\delta}$ CI contains 0). Specificity should be close to 1 (100%).

We evaluated the estimator's performance at various stages of the outbreak, defined in relation to the group's epidemic peak, i.e. the day with the highest symptom onset incidence following the first case. We hypothesise that in the early stages of an outbreak, up to the group's epidemic peak, the depletion of susceptibles is not substantial enough to significantly alter transmission dynamics. Denoting $T$ the date of the group's peak incidence, we define the *analysis time window* as the time period from the first case of the group to day $T \times \varepsilon$, where $\varepsilon$ represents any non-negative real number and is referred to as the "peak coefficient". A peak coefficient value of $\varepsilon = 1$ implies analysis until the group's peak, while values above or below 1 imply analysis using data up to before or after the peak respectively (Section 1.3 and Fig 3 in S1 File). Additionally, we introduce the term 'peak asynchronicity', calculated as the standard deviation of peak dates $T$ across groups, to measure heterogeneity in the groups' peak dates.

To assess the impact of the scenario parameters on the performance metrics, separate regressions were conducted with each performance metric as a dependent variable and

scenario parameters as independent variables (S1.4 in S1 File). These regressions provide coefficients that quantify the impact of key parameters, while the (adjusted) R-squared statistic informs on the proportion of variance explained by the model.

## Results

Fig 1 presents the estimator's performance across all epidemic scenarios considered.

Bias decreased as the analysis time window expanded, achieving near-zero levels once the group had reached its epidemic peak ($\varepsilon = 1$), with no substantial further improvements at later epidemic stages ($\varepsilon > 1$, Fig 1A).

Coverage performance was contingent upon the significance ($\alpha$) level and the stage of the group's epidemic ($\varepsilon$) (Fig 1B). Halfway before the epidemic peak (peak coefficient $\varepsilon = 0.5$), coverage at $\alpha$ levels up to 25% was too low, with average errors of 0.22, 0.18 and 0.07 for $\alpha$ levels of 5, 10, and 25%, respectively. In contrast, the 50% coverage was too high with an average error of -0.10. Around the epidemic peak ($\varepsilon$ 0.7–1.3), coverage for $\alpha = 5$–10% was good, whilst coverage for $\alpha = 25$–50% was too high (average error -0.14). At later epidemic stages ($\varepsilon$ 1.5–5),

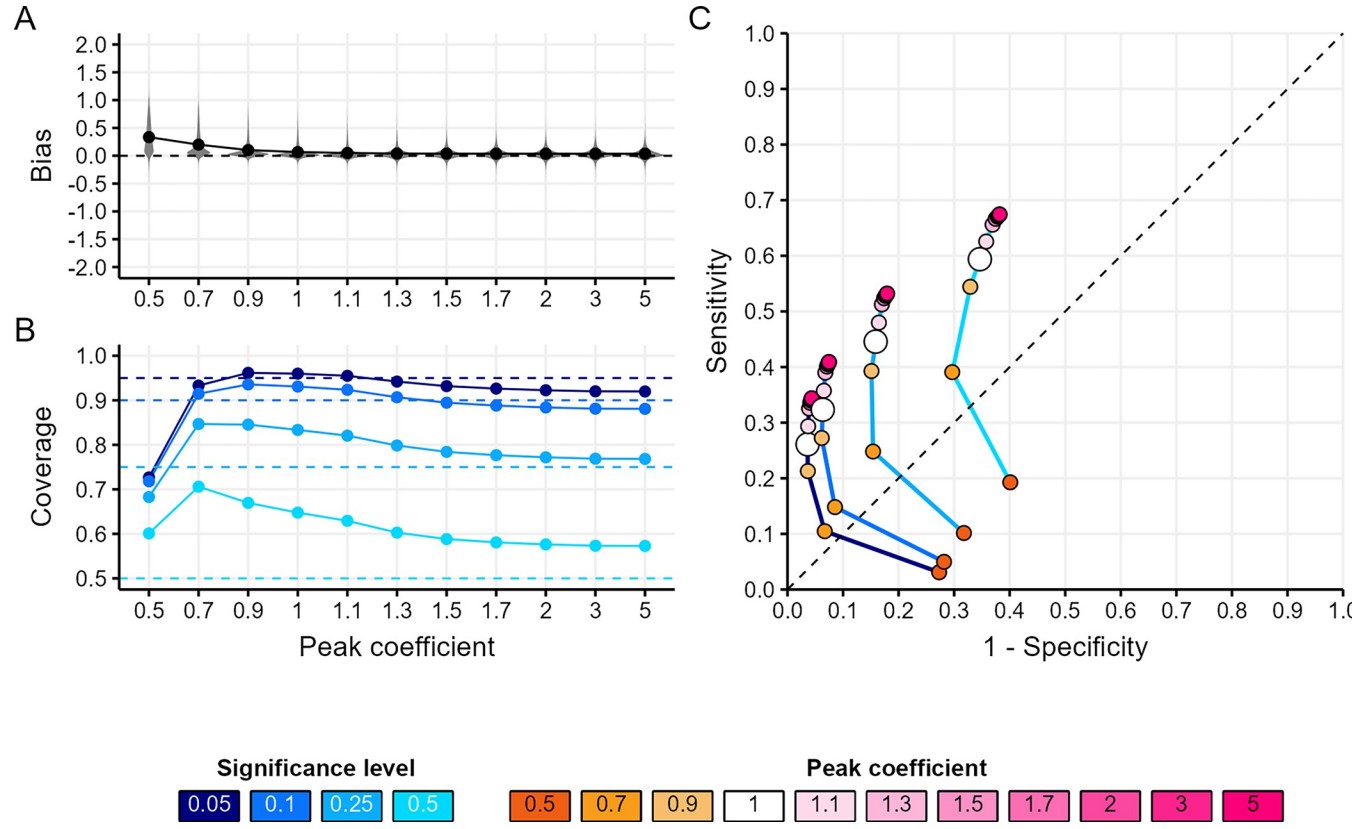

**Fig 1. Estimator's performance across all epidemic scenarios.** (A) Distribution of bias (the mean difference between the true assortativity $\delta$ value and its estimate) by peak coefficient. The peak coefficient ($\varepsilon$) is a non-negative real number used to define the *analysis time window* in relation to the group's epidemic peak. It determines the analysis period from the first case to the day $T\varepsilon$, where $T$ is the date of peak incidence for the group. A value of $\varepsilon = 1$ indicates analysis up to the group's peak date, while values above or below 1 extend the analysis to data after or before the group's peak date, respectively. The peak coefficient serves as a proxy to inform on group-level saturation, past the peak the significant depletion of susceptibles is likely to influence the underlying baseline transmission patterns. (B) Mean coverage (proportion of simulations where the true $\delta$ value is within the estimated CI) by peak coefficient for each significance level (blue shades). (C) The Receiver Operating Characteristic (ROC) (the trade-off between sensitivity and specificity) curves by peak coefficient (orange-pink points) for each significance level (blue shaded lines). In panel (A), each point shows the mean metric value across all scenarios for a given peak coefficient. In panels (B) and (C), each point shows the mean metric value across all scenarios for a given peak coefficient and significance level. Dashed lines refer to the metric's target value for (A) and (B) and represent a random classifier's ROC performance for (C).

coverage was good across most significance levels, although the 50% coverage remained high across all epidemic stages.

Sensitivity and specificity were contingent upon the CI significance level $\alpha$ and the stage of the group's epidemic ($\varepsilon$) (Fig 1C). Larger $\alpha$ values enhanced sensitivity at the expense of specificity, irrespective of the epidemic stage. And, regardless of $\alpha$, analysing transmission chains later in the epidemic (*i.e.* increasing $\varepsilon$) also enhanced sensitivity, although this improvement was marginal past a peak coefficient of 1.5. However, the gain in sensitivity relative to the loss in specificity induced by delaying the analysis varied with $\alpha$, with more pronounced tradeoffs for larger $\alpha$ values.

Fig 2 presents the relationship between various epidemic characteristics (columns) and the estimator's performance metrics (rows), for a peak coefficient of 1 and a significance level of 0.05. Additional configurations are shown in supplementary materials (Fig 6 in S1 File).

Our estimator maintained consistent unbiased performance across the entire assortativity range ($\delta$ from -1 to 1) (Fig 2 column A row 1). Coverage consistently met the 95% target for $\delta$ < 0.5, with a slight decrease in coverage performance for $\delta$ > 0.5, although coverage remained close to the target, averaging at 0.91 (sd = 0.10) (Fig 2A2). This decrease in coverage in highly assortative scenarios could be due to a saturation effect: high assortativity will accelerate the

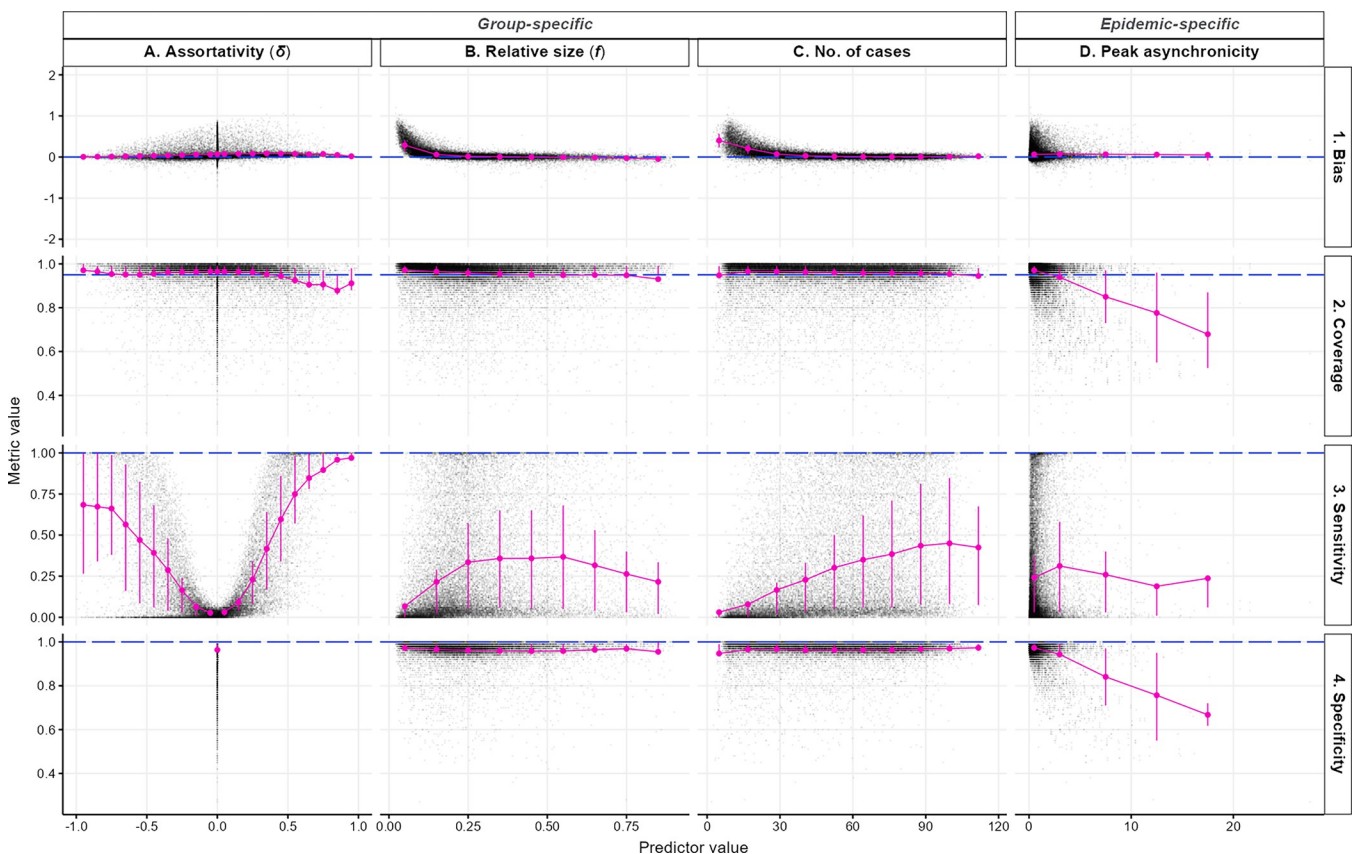

**Fig 2. Estimator's performance across scenario parameters and epidemic characteristics.** Each row corresponds to one performance indicator and each column corresponds to one simulation parameter or epidemic characteristic. In each panel, the scatter plot depicts the univariate relationship between simulation parameter or epidemic characteristic (x-axis) and the performance metric (y-axis), where each black dot represents the average observation from 100 simulations for each group in every scenario. The pink points and error bars indicate the mean and interquartile range, calculated across different bin widths: 0.1 for $\delta$ (A.) and relative group size (B.), 12.5 for the number of cases in the group (C.) and 5 days for the standard deviation of peak date (D.). Dashed blue lines indicate target metric values. Transmission chains were analysed up to the group's epidemic peak ($\varepsilon = 1$), with a significance level of 0.05.

depletion of susceptibles in the group, eventually resulting in lower observed assortativity compared to the true value (Fig 4 in S1 File). Although the assortativity coefficient $\delta$ only had a small effect on bias or coverage, it had a substantial impact on sensitivity, which was higher for larger absolute values of $\delta$. However, sensitivity rose more gradually as $|\delta|$ increased on the disassortative scale compared to the assortative scale (Fig 2A3, Table 1.1 in S1 File), reaching an average of 82% for $\delta \geq 0.5$ compared to 55% for $\delta \leq -0.5$, suggesting a better ability to detect assortative than disassortative transmission. Indeed, assortative transmission implies that transmissions propagate within the same group across multiple generations, consequently increasing the sample size ($\tau.\leftarrow a$ in Eq 6) compared to disassortative transmission, and thus narrowing the CI, thereby enhancing sensitivity. Our linear regression suggested that the assortativity coefficient explained nearly 60% of the variance observed in sensitivity (Table 1.1 in S1 File).

Increasing the number of cases substantially reduced bias (Fig 2C1, Table 2 in S1 File), and increased sensitivity (Fig 2C3, Table 1.2 in S1 File) but had little effect on specificity or coverage (Fig 2C4 and 2C2). Bias was negligible (mean: 0.04, sd: 0.07) once the group reached 30 to 40 cases. Sensitivity was positively correlated with the number of cases: controlling for $\delta$, the odds of detecting an assortative or disassortative pattern increased by 4% with each additional case (Table 1.2 in S1 File).

The relative size of the group had a substantial effect on bias (Fig 2B1, Table 2 in S1 File) and sensitivity (Fig 2B3, Table 1.2 in S1 File) but no effect on specificity (Fig 2B4) nor coverage (Fig 2B2). When groups comprised 10% or more of the total population size, bias was close to 0 (Fig 2B1), and the odds of detecting an assortative pattern increased fourfold, compared to smaller groups (odds ratios (OR) = 4.15, 95% CI = 4.07–4.24) (Fig 2B3, Table 1.2 in S1 File). Relative size and the number of cases jointly accounted for 72% of the variation in bias (Table 2 in S1 File), and contributed to a 42% increase in the pseudo R-squared for the linear regression on sensitivity (from 0.566 in Table 1.1 to 0.805 in Table 1.2 in S1 File).

Diverse transmission dynamics emerge from numerous groups, varying group sizes, reproduction numbers, and/or assortativity coefficients (Fig 5 in S1 File). This diversity results in varying saturation levels between groups over time, affecting transmission patterns within and between groups. Peak asynchronicity, a measure of heterogeneity in epidemic peak timing across groups was negatively associated with coverage (OR = 0.78, 95% CI = 0.78–0.78) and specificity (OR = 0.76, 95% CI = 0.76–0.76), explaining 18% and 24% of the variance, respectively (Tables 3 and 4 in S1 File, Fig 2D2 and 2D4). These results suggest a decrease in our estimator's performance with increasing heterogeneity between groups. However, our estimates remained unbiased (Fig 2D1) and with consistent sensitivity (Fig 2D3) irrespective of that heterogeneity.

In summary, analysing transmission chains at least up to the group's epidemic peak generally improved all performance metrics. Near the group's epidemic peak, coverage with significance levels of 5 or 10% yielded good performance, while levels of 25 and 50% were a bit too high, improving after the peak. Specificity was higher at lower significance levels, while sensitivity was higher at larger significance levels. Increased cases and relative group size contributed to improved estimator accuracy, reduced bias, and heightened sensitivity, with no significant impact on coverage nor specificity. Complex epidemic settings, measured through peak asynchronicity, did not significantly affect sensitivity or bias but were associated with a reduction in coverage and specificity.

## Discussion

We developed a method to detect and quantify the transmission assortativity of different groups based on transmission chains. We performed an extensive simulation study covering a

range of epidemic scenarios compatible with viral respiratory nosocomial outbreaks to assess the performance of our approach.

Our results indicate that the estimator's performance is influenced by assortativity patterns, relative group sizes, number of cases, and peak dates asynchronicity.

Generally, under the various settings considered—characterised by small group sizes and rapid saturation -, analysing transmission chains too early in the outbreak, before the group's epidemic peak, results in poor performance across all metrics considered. On the other hand, delaying assortativity coefficient estimation poses challenges for timely policy implementation. Choosing when exactly in the epidemic to analyse transmission chains, and what significance level to use for estimating the assortativity coefficients, will also depend on the objective. For instance, minimising bias and maximising sensitivity is best achieved later in the epidemic, past the group's peak, and using larger significance levels. Conversely, improving coverage and maximising specificity is easiest before the group's epidemic peak and using lower significance levels. Nevertheless, estimating assortativity at a target time before or at the peak requires accurate prediction of the group's peak date which can be very challenging.

As a rule of thumb, we suggest analysing all available transmission chain data up to the group's epidemic peak with a significance level of 0.05. Under this setting, our estimator provides a generally accurate measure of assortativity with reliable coverage and specificity albeit lower sensitivity.

Detecting non-homogeneous transmission patterns (sensitivity) in the presence of relatively small groups (*i.e.* a group constituting less than 10% of the total population), with groups having fewer than 30 cases is challenging, particularly when assortative or disassortative patterns are mild ($-0.5 \leq \delta \leq 0.5$). Importantly, it is considerably easier to detect assortativity than disassortativity, given that assortativity yields more transmission events within the group considered (where most new infections appear) compared to disassortativity (where new infections tend to appear in other groups, by definition). Hence, all other things being equal, larger sample sizes are more easily achieved in assortative groups.

Our approach complements traditional survey-based methods when transmission chains are available. Worby *et al.*'s relative risk estimation [2], measuring each group's proportional change in infection incidence before and after the peak, and Abbas *et al.*'s assessment method [21], comparing actual and expected proportions of infections across groups, do not consider the influence of group size. By integrating group size into our approach, we account for variations in the pool of susceptible individuals within each group, offering a more comprehensive understanding of transmission dynamics. Consequently, our approach should provide novel insights into the impact of group dynamics when estimating transmission patterns.

The main limitation of our approach pertains to the assumption that transmission chains are perfectly known. Although transmission trees can be reconstructed from data, such reconstruction effort comes with inherent uncertainty, which we have not considered here. Conventional epidemiological investigations may provide reliable transmission chains but require intensive labour for contact tracing, data collection and analysis, and may be prone to error [35]. Statistical approaches have been developed to reconstruct who infected whom using data on contacts, symptoms onset dates, and pathogen genome sequences [29], but in some contexts even these prove insufficient to precisely reconstruct transmission trees [21, 23, 36]. Our study underscores the challenges of inferring group contributions in some scenarios, even in the hypothetical instance where transmission trees are perfectly known. Nevertheless, our approach is adaptable and can be extended to reconstructed transmission chains, for example, by estimating the assortativity coefficient over all posterior transmission trees. Future research should delve into understanding how uncertainty surrounding these transmission trees further impacts our ability to infer transmission patterns.

Another limitation of our approach includes that our estimator requires, and is quite sensitive to (Fig 1.2 in S1 File), information on group sizes which may be difficult to obtain in real-life settings, however various methods exist for population size estimation [37]. Our simulations also assumed that individuals who have been infected become permanently immune, an assumption which is typically valid over short time frames but may be unrealistic over longer time horizons. Finally, characterising transmission through assortativity implies that a group's transmission patterns are identical towards all other groups, with only the within-group pattern being distinct. While this approach is fully representative in a two-group scenario, it is limiting when additional groups are involved. Nevertheless, this simplification aligns with established research on social networks and disease transmission dynamics suggesting that assortativity coefficients alone can effectively capture the essence of contact and transmission patterns across various contexts [30, 32, 38].

Despite these limitations, this study provides valuable insights into when the role of different groups in infectious disease transmission can be reliably identified in small outbreak settings, such as nosocomial outbreaks. We provide a framework for estimating group-specific transmission patterns that can be adapted to reconstructed transmission chains for real-world applications. By establishing the conditions under which these patterns become discernible, our findings can guide the timing and applicability of targeted control policies in these critical early-stage scenarios.

## Supporting information

**S1 File. Supplementary material.** Supplementary materials for the manuscript.
(DOCX)

## Acknowledgments

Simulations, analyses and visualisations were performed using the R software version 4.4.0 (https://www.R-project.org/) and the ggplot2 package (https://ggplot2.tidyverse.org/).

## Author Contributions

**Conceptualization:** Cyril Geismar, Peter J. White, Anne Cori, Thibaut Jombart.

**Data curation:** Cyril Geismar.

**Formal analysis:** Cyril Geismar.

**Funding acquisition:** Peter J. White, Anne Cori, Thibaut Jombart.

**Investigation:** Cyril Geismar, Peter J. White, Anne Cori, Thibaut Jombart.

**Methodology:** Cyril Geismar, Peter J. White, Anne Cori, Thibaut Jombart.

**Project administration:** Cyril Geismar.

**Resources:** Cyril Geismar, Peter J. White, Anne Cori, Thibaut Jombart.

**Software:** Cyril Geismar.

**Supervision:** Peter J. White, Anne Cori, Thibaut Jombart.

**Validation:** Cyril Geismar, Anne Cori, Thibaut Jombart.

**Visualization:** Cyril Geismar.

**Writing – original draft:** Cyril Geismar.

**Writing – review & editing:** Cyril Geismar, Anne Cori, Thibaut Jombart.

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
