## [Decision Letter · Decision Letter 0]

17 Oct 2024

Sorting out assortativity: when can we assess the contributions of different population groups to epidemic transmission?

PONE-D-24-35339

Dear Dr. Geismar,

We’re pleased to inform you that your manuscript has been judged scientifically suitable for publication and will be formally accepted for publication once it meets all outstanding technical requirements.

Kind regards,

Pablo Martin Rodriguez

Academic Editor

PLOS ONE

1. Thank you for stating the following financial disclosure: "CG is supported by a PhD studentship at Imperial College London funded by the National Institute for Health Research (NIHR) Health Protection Research Unit (HPRU) in Modelling and Health Economics, which is a partnership between the UK Health Security Agency (UKHSA), Imperial College London, and the London School of Hygiene & Tropical Medicine (grant code NIHR200908). AC, PJW are supported by the HPRU in Modelling and Health Economics. This work was supported by the UK Medical Research Council (MRC) Centre for Global Infectious Disease Analysis (grant number MR/X020258/1); this award comes under the Global Health EDCTP3 Joint Undertaking."

Please respond by return e-mail so that we can amend your financial disclosure and competing interests on your behalf.

Reviewers' comments:

Reviewer's Responses to Questions

**Comments to the Author**

1. Is the manuscript technically sound, and do the data support the conclusions?

Reviewer #1: Yes

2. Has the statistical analysis been performed appropriately and rigorously? 

Reviewer #1: Yes

3. Have the authors made all data underlying the findings in their manuscript fully available?

Reviewer #1: Yes

4. Is the manuscript presented in an intelligible fashion and written in standard English?

Reviewer #1: Yes

5. Review Comments to the Author

Reviewer #1: The identification of transmission patterns between and within groups may be important to understand the transmission dynamics and determine efficient measures to control epidemic outbreaks.

In this work, the authors propose a new method to estimate assortativity patterns among the distinct groups, which is based on the full knowledge of transmission chains. They also present an exhaustive simulation study that identifies the conditions under which the method performs effectively and offers guidelines for its application.

The paper is well-written, and it is noticeable that the authors have significantly improved the paper by addressing most of the questions and concerns raised by the two previous reviewers during its submission to PLOS Computational Biology.

The fact that the method requires the full knowledge is certainly an important limitation, but I tend to agree with the authors that the method can be adapted and combined with tools to probabilistically reconstruct the transmission events. Of course, this does not settle the problem but opens avenues to further research that consider uncertainty or partial observation of transmission chains.

Overall, I believe this paper provides a valuable contribution in this important topic and may establish a solid foundation for future studies. I recommend the acceptance of the paper for publication in PLOS ONE.

6. PLOS authors have the option to publish the peer review history of their article (what does this mean?). If published, this will include your full peer review and any attached files.

Reviewer #1: No

---

## [Editor Report · Acceptance letter]

22 Oct 2024

PONE-D-24-35339 

PLOS ONE

Dear Dr. Geismar, 

I'm pleased to inform you that your manuscript has been deemed suitable for publication in PLOS ONE. Congratulations! Your manuscript is now being handed over to our production team.

Kind regards, 

on behalf of

Professor Pablo Martin Rodriguez 

Academic Editor

PLOS ONE